# Detection of Selected Equine Respiratory Pathogens in Stall Samples Collected at a Multi-Week Equestrian Show during the Winter Months

**DOI:** 10.3390/v15102078

**Published:** 2023-10-11

**Authors:** Kaila Lawton, David Runk, Steve Hankin, Eric Mendonsa, Dale Hull, Samantha Barnum, Nicola Pusterla

**Affiliations:** 1Department of Medicine and Epidemiology, School of Veterinary Medicine, University of California, Davis, CA 95616, USA; kolawton@ucdavis.edu (K.L.); smmapes@ucdavis.edu (S.B.); 2Desert International Horse Park, Thermal, CA 92274, USA; drunk@sbcglobal.net (D.R.); steve@deserthorsepark.com (S.H.); 3Fluxergy, Irvine, CA 92618, USA; emendonsa@fluxergy.com (E.M.); dhull@fluxergy.com (D.H.)

**Keywords:** equine, horses, respiratory, pathogens, PCR, environment, monitoring

## Abstract

The aim of this study was to use environmental sampling to determine the frequency of detection of selected equine respiratory viruses and bacteria in horses attending a multi-week equestrian show during the winter months. At four time points during showing, environmental sponge samples were collected from all stalls on the property and tested for the presence of equine herpesvirus-1 (EHV-1), EHV-2, EHV-4, equine influenza virus (EIV), equine rhinitis B virus (ERBV), *Streptococcus equi* ss. *equi* (*S. equi*), and *S. equi* ss. *zooepidemicus* (*S. zooepidemicus*) using real-time PCR (PCR). Environmental sponges were collected from all 53 barns by using one sponge for up to 10 stalls. Further, 2/53 barns were randomly selected for individual stall sampling in order to compare the results between individual and pooled stall samples. A total of 333/948 (35.13%, 95% CI 32.09–38.26%) pooled environmental stall sponges tested PCR-positive for at least one of the selected respiratory pathogens. *Streptococcus zooepidemicus* was the most commonly detected pathogen in pooled samples (28.69%, 95% CI 25.83–31.69%), followed by EHV-2 (14.45%, 95% CI 12.27–16.85%), EHV-4 (1.37%, 95% CI 0.73–2.33%), and a very small percentage of pooled stall sponges tested PCR-positive for EHV-1, ERBV, EIV, and *S. equi*. In individual samples, 171/464 (36.85%, 95% CI 32.45–41.42%) environmental stall sponges tested PCR-positive for at least one of the selected pathogens, following a similar frequency of pathogen detection as pooled samples. The detection frequency of true respiratory pathogens from environmental samples was higher during the winter months compared to previous studies performed during spring and summer, and this testing highlights that such pathogens circulate with greater frequency during the colder months of the year. The strategy of monitoring environmental stall samples for respiratory pathogens circumvents the often labor-intensive collection of respiratory secretions from healthy horses and allows for a more efficient assessment of pathogen buildup over time. However, environmental stall testing for respiratory pathogens should not replace proper biosecurity protocols, but it should instead be considered as an additional tool to monitor the silent circulation of respiratory pathogens in at-risk horses.

## 1. Introduction

In recent years, biosecurity for equine infectious diseases has increased at large multi-week equestrian show events following concerns of respiratory outbreaks, especially equine herpesvirus 1 myeloencephalopathy (EHM) [1,2]. Vaccination requirements, health certificates, pre-show testing, daily temperature monitoring, and on-site rapid PCR testing are all examples of disease prevention protocols that have been instituted by show organizers in order to reduce the likelihood of outbreaks. However, most protocols aimed at recognizing early disease and reducing the spread rely on the willingness of riders and trainers to follow such protocols. Recent studies have shown the frequency of detection of respiratory pathogens amongst healthy horses ranging from 0% to 4% [3,4,5,6,7,8], with differences based on demographics, use of the population tested, and the time of the year. The detection of subclinical shedders, potentially linked to the spread of respiratory pathogens, represents one of the greatest challenges in the show industry [2]. The testing of individual sport horses at large equestrian events is impractical, as well as logistically and financially unsustainable. In an attempt to monitor the silent spread of respiratory pathogens, the focus has recently shifted from the individual horse to its environment [3]. While the testing of respiratory secretions in healthy sport horses gives a real-time insight into their shedding status, testing the environment determines the accumulation of respiratory pathogens over time. This approach has been successfully used to monitor the presence and potential spread of respiratory pathogens at a large, multi-week horse show during the summer months [8]. The results of that study showed that in a selected population of sport horses, the shedding frequency of respiratory viruses was low and primarily restricted to equine rhinitis B virus (ERBV) with little evidence of active transmission and environmental contamination [8]. Unfortunately, such data have yet to be established for at-risk horses during the colder months of the year, when respiratory pathogens are known to circulate with higher frequency. It was, therefore, the aim of this study to test environmental samples for selected equine respiratory pathogens in sport horses attending a multi-week equestrian show during the winter months.

## 2. Materials and Methods

### 2.1. Study Population and Location

This study was performed at a large, multi-week-long sport horse show in Thermal, CA, USA (Desert International Horse Park, deserthorsepark.com, accessed 1 October 2022) from October 2022 to March 2023. The show experienced an outbreak of EHM during their 2021/2022 season, leading to the cancelation of four show weeks. In order to reduce the risk of a subsequent outbreak and recognize diseased horses in the early stages, the show organizers instituted the following monitoring and biosecurity protocols for the 2022/2023 show season: vaccination requirements against EHV-1, health certificates, daily rectal temperature monitoring, on-site rapid EHV-1 PCR testing for febrile horses, and separation of horses with fever. Further, emphasis was placed on regular cleaning of vacated stalls, barns, and high traffic areas and in reducing direct contact between horses in and outside their respective stalls. Owners and trainers were encouraged not to share equipment between horses and to use individual buckets to provide water. 

### 2.2. Sample Collection

Environmental samples were collected from all stalls (approximately 2700 stalls) in 53 barns on the property. To reduce costs but maintain a representative sample, up to 10 stalls were swabbed with each environmental sponge. Two barns (65 stalls each) were randomly selected for sample collection from each individual stall in order to compare the results between pooled and individual stalls. Sponges were collected at four arbitrarily chosen time points: week 0 (prior to the start of showing), week 8 (before the Christmas break), week 17 (in the middle of the second period), and week 22 (at the conclusion of the 2022–2023 show season).

Stall samples were collected using individual sponges (Sponge-Stick with 10 mL neutralizing buffer, 3M, St. Paul, MN, USA). The biocide-free cellulose sponges measure 1.5 × 3 inches and are mounted at one end of a stick and prehydrated with neutralizing buffer diluent for the collection of samples. Each stall, both for pooled and individual sampling, was swabbed along the front corner where food and water buckets were kept, the inside of the stall door, and the front bars of the stall that faced the barn alleyway. For pooled sampling, gloves were changed between pooled sampled and every time a new sponge was used. For individual sampling, gloves were changed between stalls. The sponges were labeled with the date and barn and stall number and kept refrigerated and shipped on ice to the laboratory at the University of California in Davis for sample processing and analysis. 

### 2.3. Sample Analysis

The collected sponges were squeezed in order to express the neutralizing buffer. Thereafter, two hundred microliters of neutralizing buffer containing environmental samples were processed for nucleic acid extraction using a commercial extraction kit (QIAamp 96 DNA, Qiagen, Valencia, CA, USA) and an automated extraction system (QIAcube HT, Qiagen, Valencia, CA, USA) according to the manufacturer’s recommendations. Purified nucleic acid samples were tested for the presence of selected respiratory pathogens including equine herpesvirus-1 (EHV-1), EHV-2, EHV-4, equine influenza virus (EIV), equine rhinitis B virus (ERBV), *Streptococcus equi* ss. *equi* (*S. equi*), and *Streptococcus equi* ss. *zooepidemicus* (*S. zooepidemicus*) using previously reported protocols [9]. These particular pathogens were chosen to create a representational profile of pathogen shedding over time. EHV-1, EHV-4, EIV, ERBV, and *S. equi* are true respiratory pathogens commonly linked to clinical disease. They are relevant to current biosecurity concerns and cause significant clinical impact. Lesser characterized pathogens (EHV-2 and *S. zooepidemicus*) were chosen as a metric to monitor environmental contamination over time. These pathogens are more frequently detected in nasal secretions of healthy horses, but their clinical impact has remained poorly understood. Using a combination of true and lesser characterized respiratory pathogens provided a balanced overview of the pathogens present in the environment. The frequency of respiratory pathogen detection from stall samples was evaluated using descriptive analyses and the results from individual stalls and the corresponding pooled stalls was compared to determine the overall agreement.

## 3. Results

A total of 948 pooled stall sponges and 464 individual stall sponges were collected over a 22-week period. The number of pooled swabs collected at each time point ranged from 206 to 277 (median of 237) and the number of individual swabs at each time point ranged from 103 to 122 (median of 116). A total of 333 (35.13%, 95% CI 32.09–38.26%) of the pooled environmental stall sponges tested positive for at least one of the selected equine respiratory pathogens (Table 1). *Streptococcus zooepidemicus* was most commonly detected, with 272/948 sponges (28.69%, 95% CI 25.83–31.69%) testing PCR-positive. A total of 137 sponges (14.45%, 95% CI 12.27–16.85%) tested PCR-positive for EHV-2 and 13 sponges (1.37%, 95% CI 0.73–2.33%) tested PCR positive for EHV-4. A very small percentage of pooled stall sponges tested PCR-positive for EHV-1, ERBV, EIV, and *S. equi* (Table 1). The individual stall samples showed a similar pattern with a total of 171 (36.85%, 95% CI 32.45–41.42%) sponges testing PCR-positive for at least one of the selected equine respiratory pathogens (Table 2). Again, *S. zooepidemicus* was the most commonly detected pathogen with 149/464 (32.11%, 95% CI 27.88–36.57%) sponges testing PCR-positive followed by EHV-2 (11.85%, 95% CI 9.06–15.15%). EHV-1, and EHV-4 were also detected but at very low frequencies (Table 2). *S. equi,* EIV, and ERBV were not detected in individual stall sponges.

Looking at each time point separately, for both the pooled and individual samples, week 8 had the highest frequency of respiratory pathogen detection. A total of 129/333 (38.74%, 95% CI 33.48–44.20%) and a total of 87/171 (50.88%, 43.13–58.59%) of total PCR-positive samples from pooled and individual environmental sponges, respectively, were collected at this time point along with the three EHV-1 positive pooled samples. These three EHV-1 PCR-positive sponges originated from three different barns, suggesting no clustering of infection. Results from the individual stalls mirrored the results from pooled samples, with two EHV-1 PCR-positive individual samples at week 8 matching two EHV-1 PCR-positive pooled samples. The third pooled EHV-1 PCR-positive sponge did not have corresponding individual samples to compare. 

## 4. Discussion

The authors chose to test collected environmental sponges for a mixed pathogen profile. *Streptococcus zooepidemicus* and EHV-2 are lesser characterized pathogens prevalent in the nasal secretions of healthy horses and can be used to determine overall environmental contamination when horses are housed in the same environment for extended periods of time. The results from the current study showed that *S. zooepidemicus* and EHV-2 were frequently detected from the environment of healthy horses and are in agreement with a previous study showing that amongst select respiratory pathogens, *S. zooepidemicus* and EHV-2 had the highest detection frequency in nasal secretions of healthy show horses and their respective environment [10]. In contrast, EHV-1, EHV-4, EIV, ERBV, and *S. equi* are often associated with clinical disease and are, therefore, considered true respiratory pathogens. Previous studies have shown that true respiratory pathogens can be detected in nasal secretions and/or the environment of silent shedders without signs of clinical disease [3,4,5,6,7,8,10]. The detection frequency of these latter pathogens in the current study was only 2.22%. The authors have shown in previous work that respiratory pathogens can be sporadically detected from the stalls of high-risk horses during non-outbreak situations [8,10]. It is the clustering of stalls (i.e., group of adjacent stalls) testing positive for a specific contagious respiratory pathogen that indicates the silent spread and potential for a disease outbreak. The lack of environmental contamination clusters for any true respiratory pathogens and the absence of clinically diseased horses in those stalls supports the presence of silent shedders. Because of the significant financial losses caused by the 2022 EHM outbreak at DIHP, measures to prevent disease occurrence and spread focused on proper biosecurity protocols, early detection of clinical signs, and on-site EHV-1 PCR testing of febrile horses. These strategies were successful at preventing a subsequent EHM outbreak and support the use of EHV-1 environmental testing to monitor the silent spread amongst horses. A study conducted towards the end of the 2022 outbreak at DIHP detected EHV-1 by PCR in 13.8% of often-adjacent stalls housing healthy horses [11]. This is in sharp contrast to 0.32% of EHV-1 PCR-positive stalls detected in the current study. Only three pooled sponges (out of 948) tested PCR-positive for EHV-1 during week 8 and three individual sponges (out of 464) tested PCR-positive (two in week 8, one in week 17) for EHV-1 through the entirety of the testing period. Two of these individual sponges matched with the corresponding pooled stall sponges, suggesting that up to 10 stalls can be surveyed without affecting sensitivity of the testing protocol. The low frequency of EHV-1 detection and the lack of clustering in specific barns likely reflected the absence of active transmission of EHV-1. 

When comparing the present results to a similar study performed during the summer months [8], it appears that the frequency of respiratory pathogen detection in horses is driven by environmental factors; in this instance, seasonality. Various studies carried out during the colder month of the year have reported on a broader and higher detection frequency of respiratory pathogens, which generally relates to the population density and environmental persistence of respiratory pathogens [12,13,14,15]. While the population of horses was comparable between the present study and the study performed during the summer months [8], each study carried out different objectives, limiting direct data comparison. Specifically, the number of sampling time points, the targeted respiratory pathogens, and the sampling protocols differed to some extent between the two studies. However, collective observations from both studies showed that regular testing of environmental stall samples was successful at monitoring the respiratory pathogen buildup over time and indirectly assessing the silent spread. 

Although specific to the population of horses tested and the time of year that samples were collected, the results of the current study showed temporal changes in the frequency of detection of the aforementioned seven pathogens at the four sample collection time points. A concrete explanation for this observation cannot be reached; however, reasons for the differences in PCR positivity for commensal pathogens can be suggested. A general trend in the data showed a higher frequency of PCR-positive stall sponges for one or more pathogens detected at the second time point (week 8) compared to the other time points. This observation may relate to population dynamics (number and movement of horses) and environmental factors (ambient temperature and humidity). Previous studies have shown that environmental conditions affect pathogen transmission and detection frequency on various surfaces [16,17]. These studies found low pathogen detection frequency during low temperature/low humidity weather conditions. Our data, however, found that the week with the highest frequency of pathogen detection (week 8) had the lowest 5-day average humidity percentage and the second lowest average temperature of the four sampling time points. Interactions between respiratory pathogen shedding and environmental factors are still poorly understood and in need of further research. 

The small number of individual sponges collected could be considered as a limitation as it leaves a large number of stalls that were only sampled in pools and led to the inability to determine which specific stall(s) yielded PCR-positive results. However, the individual sponges collected and the subsequent mirroring of results to the pooled sponges highlight the cost effectiveness of the utilized protocol. For this facility, which is able to accommodate up to 2700 horses, it would be extremely time consuming as well as a great financial commitment to regularly collect individual samples from every stall on the property. By using pooled samples, the authors were able to test a selection of respiratory pathogens using significantly less resources. The pathogen sampling in the current study consisted only of sponges from the horse’s environment and did not include any information about pathogen detection in nasal secretions. Additional limitations of the present study include the arbitrarily chosen collection time points, as well as the inability to determine the average occupancy rate of the stalls. Even with these acknowledged limitations, environmental sampling was still successful at monitoring the selected pathogens and supplementing current biosecurity measures to avoid serious cases of infection and disease outbreaks. These data highlight the importance of clear biosecurity protocols at large equestrian events and emphasizes compliance from horse owners and trainers. The results of this study are not meant to change any biosecurity protocols, but merely show the importance of maintaining and continuing them. Sporadic testing of horse stalls during a multi-week equestrian event may be an additional step at monitoring the presence and possible silent spread of contagious respiratory pathogens. 

## 5. Conclusions

The present data showed that commensal respiratory pathogens (*S. zooepidemicus* and EHV-2) can be detected in individual stalls and pools of stalls at a high frequency. These pathogens can be used as markers to monitor environmental contamination over time. In contrast, the detection of true respiratory pathogens in the environmental stall samples was sporadic with no evidence of a silent spread. However, the detection frequency of true respiratory pathogens from environmental samples was higher during the winter months compared to previous studies performed during spring and summer, indicating that such pathogens circulate with greater frequency during the colder months of the year. It is important to keep in mind that the results relate to the location, time of the year, and specific population of show horses and should not be extrapolated to other show events and horse populations. Additional studies are highly needed so to better understand the presence and environmental buildup of respiratory viruses and bacteria in order to mitigate the risk of disease outbreaks. Although strategies such as environmental stall sampling provide an insight into the presence of contagious respiratory pathogens, they should only be used as an adjunct protocol and not as a replacement for previously well-established biosecurity protocols. 

## Figures and Tables

**Table 1 viruses-15-02078-t001:** Results of pooled environmental stall sponges tested at four different time points for selected respiratory pathogens. Each pooled sample was collected from up to 10 occupied individual stalls. A total of 53 barns and up to 2718 stalls were available at the equestrian venue.

Week	0		8		17		22		Total	
Sponges Collected	206		221		244		277		948	
Pathogen	PCR Positive	%	PCR Positive	%	PCR Positive	%	PCR Positive	%	PCR Positive	%
EHV-1	0	0%	3	1.36%	0	0%	0	0%	3	0.32%
EHV-2	9	4.37%	48	21.72%	49	20.08%	31	11.19%	137	14.45%
EHV-4	3	1.46%	4	1.81%	3	1.23%	3	1.08%	13	1.37%
EIV	0	0%	0	0%	1	0.41%	0	0%	1	0.11%
ERBV	2	0.97%	0	0%	1	0.41%	0	0%	3	0.32%
*S. equi*	1	0.49%	0	0%	0	0%	0	0%	1	0.11%
*S. zooepidemicus*	13	6.31%	110	49.77%	99	40.57%	50	18.05%	272	28.69%
Total *	21	10.19%	129	58.37%	118	48.36%	65	23.47%	333	35.13%

* Sponges testing PCR-positive for more than one respiratory pathogen were only counted once in the total positive sponge count for that time point.

**Table 2 viruses-15-02078-t002:** Results of individual environmental stall sponges tested at four different time points for selected respiratory pathogens. Two barns with 65 stalls each were randomly selected for the sampling of individual stalls.

Week	0		8		17		22		Total	
Sponges Collected	119		122		103		120		464	
Pathogen	PCR Positive	%	PCR Positive	%	PCR Positive	%	PCR Positive	%	PCR Positive	%
EHV-1	0	0%	2	1.64%	1	0.97%	0	0%	3	0.65%
EHV-2	10	8.40%	31	25.41%	10	9.71%	4	3.33%	55	11.85%
EHV-4	0	0%	1	0.82%	1	0.97%	1	0.83%	3	0.65%
EIV	0	0%	0	0%	0	0%	0	0%	0	0%
ERBV	0	0%	0	0%	0	0%	0	0%	0	0%
*S. equi*	0	0%	0	0%	0	0%	0	0%	0	0%
*S. zooepidemicus*	35	29.41%	79	64.75%	31	30.10%	4	3.33%	149	32.11%
Total *	41	34.45%	87	71.31%	36	34.95%	7	5.83%	171	36.85%

* Sponges testing PCR-positive for more than one respiratory pathogen were only counted once in the total positive sponge count for that time point.

## Data Availability

Data are available on request due to privacy restrictions.

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
