# Peer review of "Detection of Selected Equine Respiratory Pathogens in Stall Samples Collected at a Multi-Week Equestrian Show during the Winter Months"

_viruses, 2023, doi:10.3390/v15102078_

Round 1
Reviewer 1 Report
The aim of this study was to use environmental sampling to determine the frequency of detection of selected equine respiratory viruses and bacteria in horses attending a multi-week equestrian show during the winter months.
This is an important subject in view of the recent epidemics that have occurred at major equestrian events such as Valencia in 2021 and Thermal in 2022.
I would like to make some minor changes
Line 82 : Please describe the sponges : size, thickness, material, model and supplier, 3M sponge-stick (3M) ? 3M hydrated-sponge (3M)?
Line 97 : Please describe the initial volume of sample extracted with the Qiacube..
Line 97 : Please specify the exact name of the kit used on the Qiacube, QIAamp96 DNA kit ? QIAamp96 virus ? an other kit ?
Line 115 : Why does the number of samples vary from one time-point to another? Why aren't the same stalls picked over time after the initial random selection?
It would have been interesting to know whether the stalls remain positive over time, and for how long.
Line 121 : I disagree with the term qPCR used throughout the article for the stall swab. Indeed, the abbreviation qPCR describes a quantitative PCR, which requires a quantification of the viral load as described in the previous articles cited by the authors with standard curve. In this manuscript, the authors present only the positivity or negativity of samples, which is not quantitative. We only have the positive or negative status of the stalls. It would have been interesting to have an idea of the viral loads found and to see if they decrease over time. The authors could present their data in viral loads/µl of DNA extract or viral loads/ml of recovery solution of each sample. If it is not possible, remplace all « qPCR » by PCR for the results.
Table 1 and table 2 : Please review the layout of the 2 tables to make them easier to read. Example: Separate the sampling point columns with thicker vertical lines.
Author Response
Reviewer 1
The aim of this study was to use environmental sampling to determine the frequency of detection of selected equine respiratory viruses and bacteria in horses attending a multi-week equestrian show during the winter months.
This is an important subject in view of the recent epidemics that have occurred at major equestrian events such as Valencia in 2021 and Thermal in 2022.
I would like to make some minor changes
Line 82 : Please describe the sponges : size, thickness, material, model and supplier, 3M sponge-stick (3M) ? 3M hydrated-sponge (3M)?
For the purpose of the study, the investigators used 3M Sponge-Stick with 10 mL of neutralizing buffer. This 1.5 x 3 inch biocide-free cellulose sponge is mounted at one end of a stick and pre-hydrated with neutralizing buffer diluent for collection of samples. The requested information has been added under Material and Methods.
Line 97 : Please describe the initial volume of sample extracted with the Qiacube.
The initial volume of sample extraction with the Qiacube was 200 µl. The missing information was added to the manuscript.
Line 97 : Please specify the exact name of the kit used on the Qiacube, QIAamp96 DNA kit ? QIAamp96 virus ? an other kit ?
For the purpose of the study, the QIAamp96 DNA kit was used. The missing information has been added in Material and Methods.
Line 115 : Why does the number of samples vary from one time-point to another? Why aren't the same stalls picked over time after the initial random selection?
The number of samples varied because only stalls with real-time occupancy were swabbed.
It would have been interesting to know whether the stalls remain positive over time, and for how long.
The reviewer brings up a very interesting aspect of the study. It would have been interesting to determine how long the same individual and pooled stalls stayed positive for the selected pathogens. Because of the rare detection of common respiratory pathogens, all the positive individual stalls and stall pools represented a single event. Unfortunately because of the study design, this information could not be retrieved for the lesser-characterized respiratory pathogens.
Line 121 : I disagree with the term qPCR used throughout the article for the stall swab. Indeed, the abbreviation qPCR describes a quantitative PCR, which requires a quantification of the viral load as described in the previous articles cited by the authors with standard curve. In this manuscript, the authors present only the positivity or negativity of samples, which is not quantitative. We only have the positive or negative status of the stalls. It would have been interesting to have an idea of the viral loads found and to see if they decrease over time. The authors could present their data in viral loads/µl of DNA extract or viral loads/ml of recovery solution of each sample. If it is not possible, remplace all « qPCR » by PCR for the results.
The investigators agree with the reviewer that the PCR results were not quantitative, therefore the term of qPCR should not be used. Instead, the investigators propose to use the term real-time PCR (PCR).
Table 1 and table 2 : Please review the layout of the 2 tables to make them easier to read. Example: Separate the sampling point columns with thicker vertical lines.
As suggested by the reviewer and in order to facilitate the reading of the table, the sampling point columns have been separated with ticker vertical lines.
Reviewer 2 Report
The manuscript by Lawton et al. is well-written and conducted, with a relevant contribution to the advances in the knowledge of the epidemiology of some equine respiratory viruses and bacteria. I support its further processing after appropriate modifications as outlined below:
Considering the type of the manuscript – Communication - The title, abstract, and introduction are well-written and effectively convey the context and importance of the study.
L15: to be more informative for the reader, please mention the meaning of the used abbreviation for the targeted diseases
L19: when you express overall prevalence values, please indicate in brackets the values of the 95% confidence interval
The "Materials and Methods" section is comprehensive, ensuring the study's reproducibility and offering a clear understanding of the techniques used. However, it would be important for the reader if the authors could justify their choice in the processed total number of samples. Was previously an estimated prevalence value taken into consideration? In this regard, I suggest that the authors refer to a statistical model, based on which they can validate the study results. So, the authors must convince the scientific community that the results are completely supportable by statistical tools, without any doubt of speculation. The random selection strategy is incomplete.
The alignment of the expected findings with the actual results adds credibility to the research.
Sentences from lines 115-117 present materials and methods data, rather than results.
The Tables 1 and 2 must be presented following the requested designs by MDPI journals
The discussion section effectively interprets the results and provides a deeper understanding of their significance.
L236-240: These sentences from the conclusion section must be rejected. They present general information and not derived conclusions from the present study.
The conclusion section must be more concise presenting also the study limitations and further strategies in the approached research area.
The authors declared: “Institutional Review Board Statement: Not applicable. And Informed Consent Statement: Not applicable” From my point of view this is a questionable statement.
Author Response
Reviewer 2
The manuscript by Lawton et al. is well-written and conducted, with a relevant contribution to the advances in the knowledge of the epidemiology of some equine respiratory viruses and bacteria. I support its further processing after appropriate modifications as outlined below:
Considering the type of the manuscript – Communication - The title, abstract, and introduction are well-written and effectively convey the context and importance of the study.
L15: to be more informative for the reader, please mention the meaning of the used abbreviation for the targeted diseases
The abbreviations for the various targeted pathogens have been spelled out.
L19: when you express overall prevalence values, please indicate in brackets the values of the 95% confidence interval
All confidence intervals have been calculated and added in the abstract as well as the results.
The "Materials and Methods" section is comprehensive, ensuring the study's reproducibility and offering a clear understanding of the techniques used. However, it would be important for the reader if the authors could justify their choice in the processed total number of samples. Was previously an estimated prevalence value taken into consideration? In this regard, I suggest that the authors refer to a statistical model, based on which they can validate the study results. So, the authors must convince the scientific community that the results are completely supportable by statistical tools, without any doubt of speculation. The random selection strategy is incomplete.
The alignment of the expected findings with the actual results adds credibility to the research.
Because this is the first study of its kind, no sample power calculation was performed. However, one needs to consider that the data is very specific to the location, population of horses present during the collection period and time of the year. In order to minimize the risk of a bias towards a certain number of selected stalls, the investigators decided to collect environmental samples from all the occupied stalls at the 4 different time points.
Sentences from lines 115-117 present materials and methods data, rather than results.
The investigators are not quite sure if the issue refers to the first sentence of the results, i.e. “A total of 948 pooled stall sponges and 464 individual stall sponges were collected over a 22-week period”. If this is the sentence of concerns, the investigators would like to highlight that this is the first time the total amount of collected samples is being mentioned. If deemed necessary, the sentence can easily be moved to Material and Methods.
The Tables 1 and 2 must be presented following the requested designs by MDPI journals
The discussion section effectively interprets the results and provides a deeper understanding of their significance.
In agreement with the design of MDPI journals and also at the request of reviewer #1, the format of the Table 1 and Table 2 has been changed.
L236-240: These sentences from the conclusion section must be rejected. They present general information and not derived conclusions from the present study.
The conclusion section must be more concise presenting also the study limitations and further strategies in the approached research area.
The first sentence of the conclusion was removed and the conclusion was changed in order to be more concise, state the study limitations and future strategies in the research field.